# High-Affinity Antibodies Designing of SARS-CoV-2 Based on Molecular Dynamics Simulations

**DOI:** 10.3390/ijms24010481

**Published:** 2022-12-28

**Authors:** Zihui Tian, Hongtao Liu, Shuangyan Zhou, Zengyan Xie, Shuai Yuan

**Affiliations:** Chongqing Key Laboratory of Big Data for Bio Intelligence, Chongqing University of Posts and Telecommunications, Chongqing 400065, China

**Keywords:** SARS-CoV-2, antibody design, MM-GBSA, fixed-point mutation

## Abstract

SARS-CoV-2 has led to a global pandemic of new crown pneumonia, which has had a tremendous impact on human society. Antibody drug therapy is one of the most effective way of combating SARS-CoV-2. In order to design potential antibody drugs with high affinity, we used antibody S309 from patients with SARS-CoV as the target antibody and RBD of S protein as the target antigen. Systems with RBD glycosylated and non-glycosylated were constructed to study the influence of glycosylation. From the results of molecular dynamics simulations, the steric effects of glycans on the surface of RBD plays a role of “wedge”, which makes the L335-E340 region of RBD close to the CDR3 region of the heavy chain of antibody and increases the contact area between antigen and antibody. By mutating the key residues of antibody at the interaction interface, we found that the binding affinities of antibody mutants G103A, P28W and Y100W were all stronger than that of the wild-type, especially for the G103A mutant. G103A significantly reduces the distance between the binding region of L335-K356 in the antigen and P28-Y32 of heavy chain in the antibody through structural transition. Taken together, the antibody design method described in this work can provide theoretical guidance and a time-saving method for antibody drug design.

## 1. Introduction

Severe acute respiratory syndrome coronavirus 2 (SARS-CoV-2) has caused widespread infections worldwide since December 2019 and caused approximately 633 million infections and 6.6 million deaths worldwide as of 10 November 2022 [1]. Although many drugs have entered clinical trials, the immune escape caused by the high mutation rate of SARS-CoV-2 makes it hard to completely overcome the virus.

SARS-CoV-2 is a single-stranded RNA virus and is a new member of the coronavirus family, with a length of 29 kb [2]. It is highly infectious with low immunity. High infectiousness is the main cause of the SARS-CoV-2 pandemic. During the infection of SARS-CoV-2, the binding of spike glycoprotein (S protein) of SARS-CoV-2 to the angiotensin converting enzyme2 (ACE2) in host cells is the first and critical step [3]. S protein exists mainly as a trimer in vivo with its highly glycosylated state. The monomer of S protein can be divided into the S1 subunit and the S2 subunit. The S1 subunit includes the N-terminal structural domain (NTD) and the receptor binding domain (RBD). For the RBD region, it is the reported key domain for the binding of S protein to ACE2. The S2 subunit contains the fusion peptide region (FP), the heptapeptide repeat sequence region HR1 and HR2, the transmembrane region (TM) and the cytoplasmic structural domain (CP) [4]. The cartoon model of the structure of SARS-CoV-2 is shown in Figure 1, which was drawn with BioRender on 30 June 2022 (https://app.biorender.com/biorender-templates). Among them, the RBD is the main epitope of SARS-CoV-2 that induces neutralizing antibody and is usually used as the main antigenic target for drug design [5].

Presently, the main treatment strategy for novel coronavirus pneumonia (COVID-19) can be divided into small-molecule drugs and antibody drugs [6]. Since small molecules cannot be specifically recognized, they are only effective in patients with mild symptoms, not in critically ill patients. In contrast, antibodies with a specific recognition function can prevent the infection of SARS-CoV-2 by competitively binding to the RBD, and thus blocking the binding of the RBD to human ACE2 [7,8]. Therefore, using antibodies for the treatment of patients with severe COVID-19 is a promising strategy.

Sequence comparison revealed that the sequence similarity between SARS-CoV-2 and SARS-CoV reached 79.6%, and the two were highly homologous [9]. Therefore, screening for antibodies resistant to SARS-CoV-2 from SARS-CoV patients is also a good choice. Antibody S309 is an antibody extracted from patients infected with SARS-CoV [10]. It targets the same conserved epitopes of SARS-CoV and SARS-CoV-2; therefore, S309 has neutralizing activity against both and should prevent immune escape caused by viral mutations.

To effectively prevent and treat SARS-CoV-2, it is still urgent to design antibodies with high binding affinity against SARS-CoV-2. The high mutation rate and low immunogenicity make the infection rate of SARS-CoV-2 high, and the high infection rate puts researchers who conduct physiological experiments at risk of being infected. Therefore, the design and screening of antibodies with high-affinity by physiological experiments involves the risk of infection and should be supported by strict experimental conditions and great expense. In addition, specific antibodies can be extracted directly from patients, but the affinity of antibodies for viruses is usually low. Therefore, this study aims to investigate the interaction mechanism between the RBD of SARS-CoV-2 and antibody S309 through molecular dynamics simulations and to modify S309 to design antibodies with high affinity. This work should provide a theoretical basis for the design of high-affinity antibodies.

## 2. Results

### 2.1. Polysaccharide Stabilizes the Interaction of Antigen and Antibody

To determine the role of glycans in antigen–antibody binding, molecular dynamics simulations were performed for the GLYCAN (with RBD glycosylated) and NO-GLYCAN (without glycosylation of RBD), respectively [11]. By calculating the RMSD values of the Cα atoms referring to the initial RBD region, it can be found that the GLYCAN system reaches equilibrium after 25 ns, and the NO-GLYCAN system reaches equilibrium after 80 ns, as shown in Figure 2A. The RMSD is significantly larger than that of the GLYCAN system. Calculation of the RMSF values of the two systems revealed that the RMSF values of most residues in the antigenic RBD region of GLYCAN were lower than those of NO-GLYCAN, indicating that most residues in the RBD region of GLYCAN were less flexible than those of NO-GLYCAN, as shown in Figure 2B. Only the RMSF of regions L335-E340, N440-N450, and P491-T500 are close to those of NO-GLYCAN. This may largely be attributed to the antigen–antibody interaction at this region. That is, the antigen–antibody interaction at this region reduced the structural flexiblity. From the above analysis, we believe that the glycans stabilize the antigen–antibody complex structure.

To evaluate the binding affinity of the two systems, the binding free energy of antigen–antibody interaction was calculated by Ambertools [12], as shown in Table 1. It was found that the binding free energy between RBD and S309 in GLYCAN was −58.6168 kcal/mol and that of NO-GLYCAN was −41.2909 kcal/mol. The calculated binding affinities further verify the important role of glycans in stabilizing the binding of antigen and antibody.

Furthermore, to investigate the key residues in antigen–antibody interactions, We performed energy decomposition of the binding free energy, as shown in Figure 3. We found that the energy contribution of key residues in the GLYCAN system was generally decreased in the NO-GLYCAN system. The results revealed that the key residues of interacting antigens were mainly distributed in the regions of T333-K356 and N439-L440, including residues N334, L335, P337, N343, A344, T345, R346, N439 and L440. The key residues of S309 are mainly distributed in the CDR1 region of P28-Y32 and the CDR3 region of Y100-I111 in the heavy chain. Key residues in the antibody are located in the heavy chain, including P28, Y100, G103, A104, W105, F106, G107, S109, L110 and I111. Therefore, we speculate that this part of the residues has an important role in glycan-promoted interactions.

Since hydrogen bonding (H-bond) plays an important role in antigen–antibody binding, to investigate the interaction between antigen and antibody we performed hydrogen bonding network analysis [13], as shown in Table 2. The results revealed that the H-bond interactions in the NO-GLYCAN showed a decreasing trend. The occupancy of H-bond in the NO-GLYCAN system is significantly lower than that of GLYCAN, especially for H-bonds of N343-Y100 and G107-E340, with the former occupancy decreasing from 41.33% to 2.9% and the latter decreasing from 39.33% to 17.88%. As shown in Figure 4, there is an obvious torsion for the aromatic ring of Y100 in the NO-GLYCAN, which makes it difficult to form H-bond with N343. While for the H-bond G107-E340, a slight angular deflection was also observed for the side-chain of E340 in the NO-GLYCAN, explaining the reduced H-bond occupancy of G107-E340.

Based on these findings, we further performed solvent-accessible surface area calculations for the hot-spot residues. We found that the major changes in solvent-accessible surface area were concentrated on residues Y32 and Y54, which were located in the heavy chain of the antibody. For residue Y32, the solvent accessible surface increased from 577.9995 Å${}^{2}$ in GLYCAN to 682.484 Å${}^{2}$ in NO-GLYCAN. For residue Y54, it increased from 554.0865 Å${}^{2}$ in GLYCAN to 649.0179 Å${}^{2}$, as shown in Figure 5. The increased solvent-accessible surface represents an increase in the contact area of the probe water molecule with the antigen surface, which means that the area of antigen–antibody contact will be reduced. Therefore, we hypothesize that the presence of glycans causes residues Y32 and Y54 of the heavy chain to be structurally closer to the antigen, resulting in a tighter binding to the antigen and thus a change in the solvent-accessible surface area.

Based on the above results, we speculate that the presence of glycans causes a large change in the conformation of the GLYCAN, and this change facilitates antigen–antibody binding. To verify this speculation, we performed a radius of gyration analysis for GLYCAN and NO-GLYCAN, as shown in Figure 6. The radius of gyration of NO-GLYCAN is larger than that of GLYCAN, indicating that the structure of GLYCAN is more compact than that of NO-GLYCAN, and the binding between antigen and antibody is tighter.

In order to investigate the difference of the structural changes between the GLYCAN and the NO-GLYCAN, we further performed a clustering analysis based on the equilibrium trajectories (80–100 ns), and the obtained representative conformations were used to perform a superposition analysis, as shown in Figure 7. We found that on the antigen–antibody contact surface, partial residue of CDR3 in the heavy chain of GLYCAN was closer to the RBD region of L335-E340 than that of NO-GLYCAN. Among them, Y100-T103 on CDR3 and L335-E340 on RBD have the most structural changes in the region, as shown in Figure 8A. Since the glycan is located at the interaction surface of the RBD with the heavy chain of antibody and is directly linked to the residue N334 of RBD (Figure 7), we believe that the steric hindrance of RBD surface glycans plays the role of “wedge”. The steric hindrance brings the L335-E340 residue region of RBD structurally closer to the CDR3 region of the heavy chain [14,15]. L335-E340 undergoes a strong hydrogen bonding interaction network with residues of CDR3. In Figure 8B, L335-E340 is surrounded by CDR3 and residues in CDR3 form five hydrogen bonds with E340. This also increases the contact surface of residues in the CDR3 region with the RBD, which allows for better interaction with the antibody. While in the NO-GLYCAN system, an obvious twist of Y100 is observed, and the twist of Y100 causes the L335-E340 to no longer be surrounded by CDR3 (Figure 8C). Moreover, the H-bonds between E340 and G107, E108 were broken. Taken together, the glycans at the surface of RBD can enhance the interaction between antibody and RBD of the S protein.

### 2.2. Antibody Mutation Design

#### 2.2.1. Single Mutant Screening

Glycans can enhance antigen–antibody interactions. In order to obtain antibodies with stronger affinity to SARS-CoV-2, we performed mutation on key residues on the glycosylated S309. Single mutation scanning of key residues and stability evaluation of mutated antibodies were performed by using the Cartesian_ddG module in Rosetta software [16]. Only the key residues P28, D99, Y100, G103, A104, W105, G107 and L111 on the heavy chain of the antibody were mutated into another 19 amino acids one by one, since the key residues can largely affect the binding affinity. The method is able to improve the efficiency of antibody scanning and ensure the accuracy as well. The results are shown in Table 3, and the mutant systems with energy changes less than −3 kcal/mol were presented. Less than −3 kcal/mol means the mutant structure is more stable [17]. We obtained P28W, G103A, Y100W, D99T mutant system by screening.

#### 2.2.2. Single Mutant Affinity Simulation Validation

In order to further validate the predicted results, we performed molecular dynamics simulations on the mutant antibodies P28W, G103A, Y100W and D99T [18,19]. From the RMSD curves in Figure 9A, we can find that each system reached equilibrium after 40 ns in the RMSF analysis (Figure 9B–D). By comparing the low peaks with those of WT, we found that the residue region of low peak for the Y100W mutant antibody was almost unchanged, and only the value of the P28-Y32 region was decreased. In the rest of the mutant antibodies, all of them were reduced, with the G103A mutant showing the largest change.

To further confirm the reliability of the mutation prediction results, we further evaluated the binding free energy between antigen and the designed antibodies. The binding free energy was calculated by the MM-GBSA method. We found that among the four mutant systems, only the binding free energy of mutant Y100W was weaker than that of the wild-type, while the binding free energy of the rest of the mutants was stronger than that of the wild-type WT, especially for the antibody of the G103A mutant, as shown in Table 4. This means that the neutralization ability of SARS-CoV-2 was significantly enhanced in most of the mutant systems except for Y100W, among which G103A had the most significant effect.

#### 2.2.3. Representative Conformational Superposition Analysis

To verify the structural changes, we performed a cluster analysis of the equilibrium trajectories (70–100 ns) for each system to obtain the representative conformation of each system. Among the superimposed conformations of each mutant system and the wild-type system, we found that only mutant G103A showed significant changes in the antigen–antibody binding surface compared to the structure of WT, including segment L335-K356 of the antigen and P28-Y32 of the heavy chain of the antibody, as shown in Figure 10. By calculating the distance of antigen L335-K356 and the heavy chain of antibody P28-Y32 [20,21], it can be found that the distance between antigen and antibody residues of mutant G103A was significantly reduced, with the most significant changes in E340 in the RBD region and S31 and Y32 in the heavy chain of the antibody, which explained why hydrogen bonding interactions were enhanced in this region.

## 3. Discussion

The binding free energy of the GLYCAN was significantly greater than that of the NO-GLYCAN, suggesting that glycans can enhance antigen–antibody interactions. The analysis of solvent-accessible surface area, radius of gyration and hydrogen bonding network reveals the mechanism of enhanced antigen–antibody interaction by the glycan. Since the glycan is located at the interaction surface of the RBD and the heavy chain of the antibody, we suggest that the steric effects of the glycan on the surface of the RBD plays the role of a “wedge”, which brings the L335-E340 residue region of the RBD closer to the CDR1 region of the heavy chain of the antibody, resulting in a stronger hydrogen bonding network with the CDR3 residue [22,23].

In the antibody mutation design, we chose GLYCAN as the wild-type system. In mutant studies, we first eliminate a large number of mutant sites and mutant residue options that do not have mutational potential by means of prediction. This significantly reduces the time and cost of antibody design. The predicted results are further analyzed and verified by molecular dynamics simulation, thus ensuring the accuracy of the results. This method is suitable for both large-volume and small-scale mutation design.

Diamond’s team has verified through mouse experiments that S309 has mechanisms of protection against Omicron variants [24]. Meanwhile, Minh Tuyet Ma’s team found through biological experiments that S309-CAR-NK has a better affinity than other antibodies in chimeric antigen receptor (CAR)-natural killer (NK) immunotherapy and has fewer side effects [25]. These biological experiments confirm the immune potency of S309 against Omicron. Therefore, our design of antibody S309 will also be a reference for Omicron’s control.

The results show that the antibody design and screening method in this paper is theoretically feasible. Our designed antibody G103A has high binding affinity for SARS-CoV-2. Due to the limitation of computational resources, we designed only a single point mutation, and we believe that a combinatorial mutation of antibody S309 may further improve its binding affinity for SARS-CoV-2.

## 4. Materials and Methods

### 4.1. Molecular Dynamics Simulation

The crystal structure of S309 bound to the S protein of SARS-CoV-2 was obtained from the Protein Data Bank (PDB ID: 6WPT). The other part in the complex was removed, retaining only the structure of antibody S309 and RBD. In order to investigate the role of glycans in the binding of antigen–antibody, we kept the glycans in the structure to build the glycosylated type (GLYCAN) and removed the glycans to build the sugar-free type(NO-GLYCAN). All systems were described by the CHARMM 36 force field [26]. A rectangular water box with a protein-to-box edge of 10 Å was used to mimic the solvent environment; 0.15 mol/L NaCl was added to the water box to satisfy the physiological environment. All simulations were performed by using NAMD software with thermodynamic temperature set to 303 K [27]. At the first step, water molecules and counterions were relaxed by restraining the complex with a harmonic constant of 2.0 kcal/mol·Å${}^{-2}$ based on the steepest descent method. At the second step, the restraint was removed to allow all of the atoms to move freely using the conjugate gradient algorithm. After that, each system was gently heated from 0 to 303 K in 500 ps at constant volume with a harmonic constant of 10.0 kcal/mol·Å${}^{-2}$ and then equilibrated at 303 K and 1 bar constant pressure for 1000 ps. Finally, a 100 ns MD simulation was performed with a time step of 2 fs. During the simulation, all bonds involving hydrogen atoms were constrained using the SHAKE algorithm. The non-bonded cutoff was set to 10.0 Å, and electrostatic interactions were calculated using the particle-mesh Ewald method (PME). The temperature was controlled using the Langevin thermostat method [28].

### 4.2. MM-GBSA Calculation

MM-GBSA [29] is one of the most widely used methods for free energy prediction calculations [30]. Snapshots extracted from the equilibrium trajectory (80–100 ns, 3000 frames) were used for the binding free energy calculation. The solvent-accessible surface area (SASA) was estimated by the MSMS algorithm with a probe radius of 1.4 Å, the generalized Born method was used with igb set to 2; saltcon, which is salt concentration, can be set to 0.1 [31]. The formula for the free energy calculation is as follows:(1)$$\Delta G\_bind=G_{complex}-G_{antigen}-G_{antibody}$$
(2)$$\Delta G\_bind=\Delta E_{MM}+\Delta G_{GB}+\Delta G_{SA}-T\Delta S$$
(3)$$\Delta G\_bind=\Delta E_{vdw}+\Delta G_{ele}+\Delta G_{GB}+\Delta G_{SA}-T\Delta S$$
where Δ*E*${}_{MM}$ is the gas-phase intermolecular interaction energy, which contains Δ*E*${}_{vdw}$ and Δ*G*${}_{ele}$, Δ*G*${}_{GB}$ and Δ*G*${}_{SA}$ represent the polar solvation and nonpolar solvation effects, respectively, and *T*Δ*S* represents the entropy change. In this study, the contribution of the conformational entropy was not considered because the calculation of this term usually brings large errors for large-size systems such as those studied in this work [32], which do not improve the calculation accuracy of the binding free energy [33].

In order to obtain the key residues for the interaction between antigen and antibody, we also performed energy decomposition. This can provide critical information about the local dominant interactions between the amino acids in RBD and ACE2, especially those located at the interface. The calculated binding free energy was decomposed to each amino acid residue to obtain the key residues in antigen–antibody interactions.

### 4.3. Residue Mutation Prediction

We used the Cartesian_ddG [34] panel in Rosetta for single mutation scanning and stability prediction of modified antibodies. The prediction algorithm uses Cartesian space optimization to allow for small movements of the protein backbone, making the accuracy higher than that of similar prediction algorithms [35,36]. Using this method, it is possible to exclude a large number of sites and residues that do not increase binding affinity.

## 5. Conclusions

Specific recognition of antibodies is one of a promising way to treat SARS-CoV-2, so the selection of antibodies is important. We found that with the RBD glycosylated, the contact areas between residues Y32 and Y54 on the heavy chain of antibody S309 and the antigen RBD were increased; the hydrogen bonding between Y100 and G107 on the CDR3 of S309 and N343, E340 of the antigen RBD were enhanced, thus improving the antigen–antibody binding ability. Mutant G103A enhanced the hydrogen bonding interaction network between E340 of antigen RBD and S31 and Y32 on antibody S309 heavy chain CDR1 by structural alteration. This resulted in a substantial enhancement of the neutralization ability of mutant G103A against SARS-CoV-2. In the selection of antibodies against SARS-CoV-2 mutants, G103A can be considered as a potential antibody to fight against SARS-CoV-2 mutants. Meanwhile, the precise antibody design method described in this paper can be considered.

## Figures and Tables

**Figure 1 ijms-24-00481-f001:**
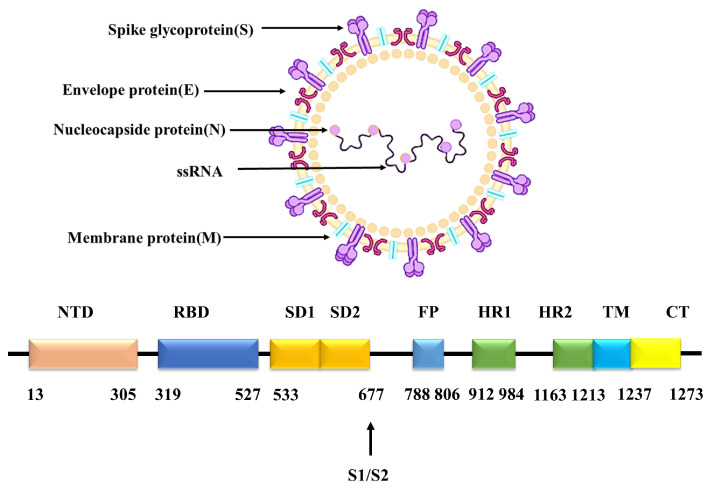
SARS-CoV-2 structure.

**Figure 2 ijms-24-00481-f002:**
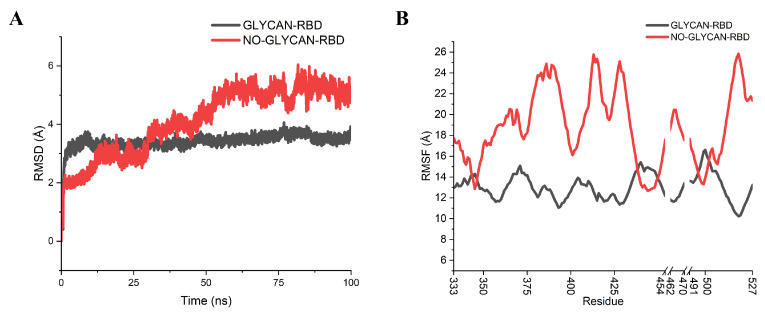
RMSD, RMSF analysis of GLYCAN and NO-GLYCAN RBD regions. GLYCAN system in black; NO-GLYCAN system in red; (**A**) RMSD analysis of antigenic RBD; (**B**) RMSF of antigenic RBD.

**Figure 3 ijms-24-00481-f003:**
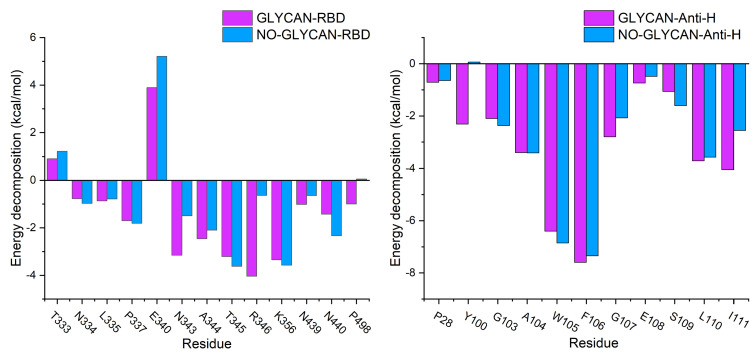
Free energy decomposition of key residues. Purple column is GLYCAN; blue column is NO-GLYCAN.

**Figure 4 ijms-24-00481-f004:**
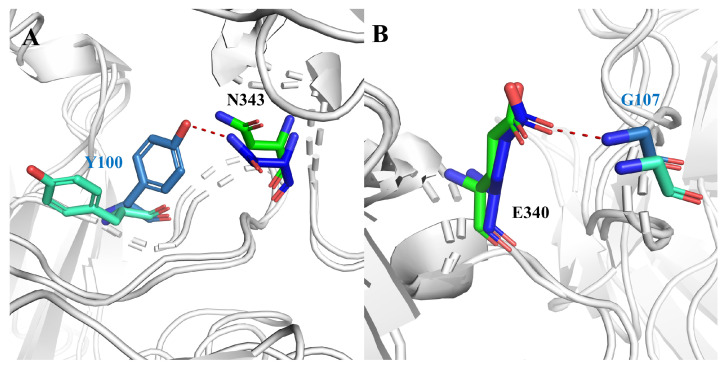
Hydrogen bond changes of GLYCAN and NO-GLYCAN: (**A**) hydrogen bond N343-Y100; (**B**) hydrogen bond G107-E340; blue for GLYCAN antigen RBD; skyblue for GLYCAN antibody S309; green for NO-NOGLYCAN antigen RBD; blue-green for NO-GLYCAN antibody S309.

**Figure 5 ijms-24-00481-f005:**
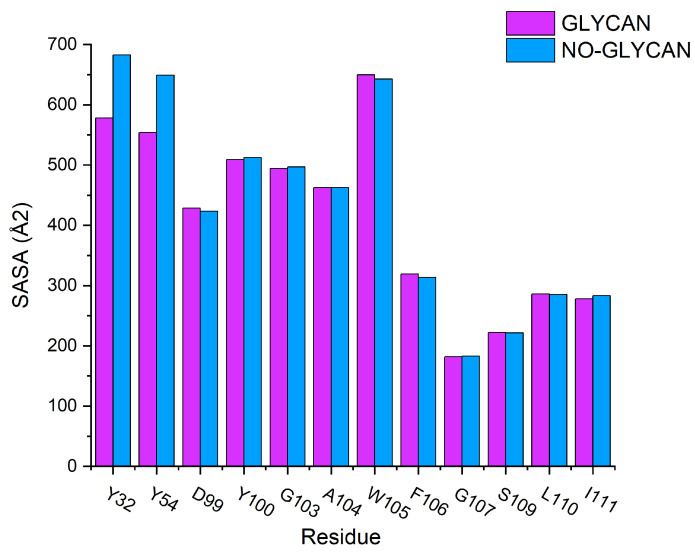
Calculation of solvent accessible surface area: GLYCAN for the purple column and NO-GLYCAN for the blue column.

**Figure 6 ijms-24-00481-f006:**
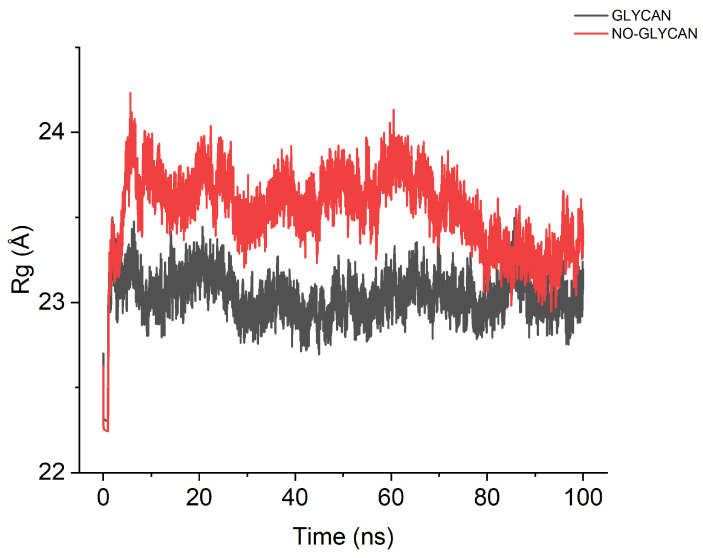
Calculation of the radius of rotation. The black curve is GLYCAN; the red curve is NO-GLYCAN.

**Figure 7 ijms-24-00481-f007:**
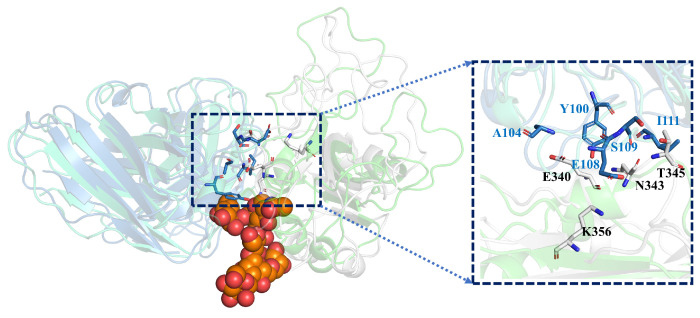
The wedge effect of polysaccharide: gray for GLYCAN antigen RBD; skyblue for GLYCAN antibody S309; green for NO-GLYCAN antigen RBD; blue-green for NO-GLYCAN antibody S309; orange for GLYCAN system of polysaccharide; boxes are antigen–antibody binding surfaces.

**Figure 8 ijms-24-00481-f008:**
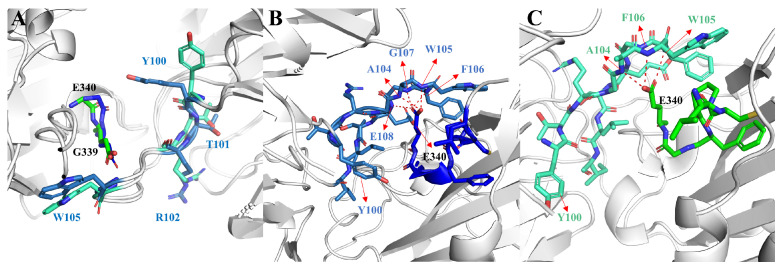
Changes in structure and hydrogen bond network of key residues in GLYCAN and NO-GLYCAN: (**A**) the structural difference of key residues located at the antigen–antibody interaction interface; (**B**) the hydrogen bond network between L335-E340 and CDR3 residues in the GLYCAN; (**C**) the hydrogen bond network between L335-E340 and CDR3 residues in the NO-GLYCAN; blue for GLYCAN antigen RBD; skyblue for CLYCAN antibody S309; green for NO-GLYCAN antigen RBD; blue-green for NO-GLYCAN antibody S309.

**Figure 9 ijms-24-00481-f009:**
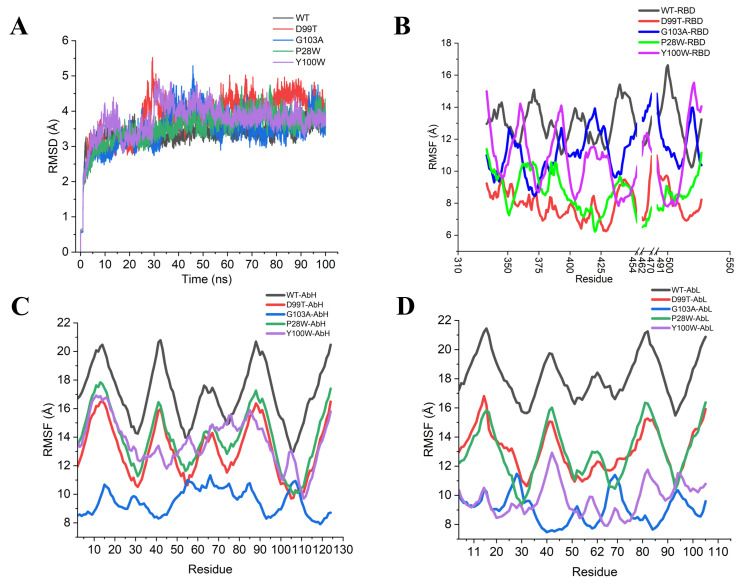
RMSD and RMSF analysis of wild-type and mutant type: (**A**) RMSD analysis; (**B**) RMSF analysis of residues in the RBD region; (**C**) RMSF analysis of residues in the heavy chain of the antibody; (**D**) RMSF analysis of residues in the light chain of the antibody.

**Figure 10 ijms-24-00481-f010:**
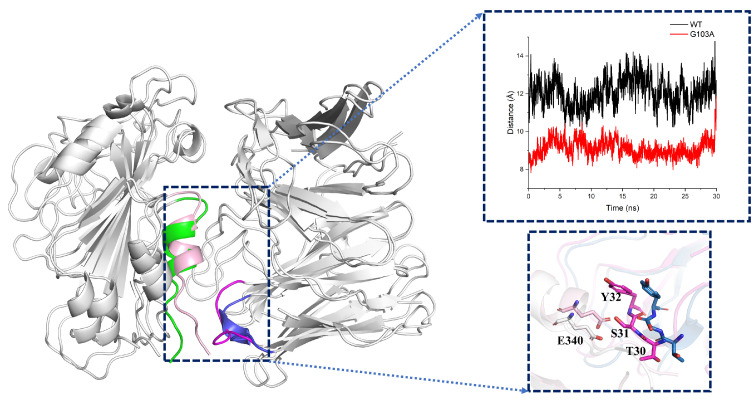
Distance and conformational changes of key residues on the binding surface of wild-type and mutant type: wild-type: green is the region of RBD L335-K356 residues, blue is the region of the heavy chain of antibody P28-Y32 residues; mutant type: light pink is the region of RBD L335-K356 residues, magenta is the region of the heavy chain of antibody P28-Y32 residues (for easy observation, all glycans of the system have been hidden in the figure). The graph shows the distance between the center of the mass of antigen residues L335-E340 and antibody residues P28-Y32; black is the WT system; red is the G103A system; and the input trajectory file is the 30 ns equilibrium trajectory of WT, G103A and D99T.

**Table 1 ijms-24-00481-t001:** Binding free energy of GLYCAN and NO-GLYCAN.

	GLYCAN (kcal/mol)	NO-GLYCAN (kcal/mol)
ΔE${}_{vdw}$	−73.279	−74.057
ΔE${}_{ele}$	−239.955	−107.104
ΔG${}_{GB}$	270.4689	150.9991
ΔE${}_{surf}$	−12.0327	−10.9644
ΔE${}_{gas}$	−313.234	−181.161
ΔG${}_{sol}$	258.4362	140.0348
ΔG${}_{bind}$	−54.7977	−41.1265

**Table 2 ijms-24-00481-t002:** Hydrogen-bonding occupancies between antibody and antigen of GLYCAN and NO-GLYCAN.

Donor	Acceptor	Occupancy	
		GLYCAN	NO-GLYCAN
ILE111-Main	ASN343-Main	48.44%	35.66%
ALA104-Main	GLU340-Side	78.33%	87.01%
TRP105-Main	GLU340-Side	78.89%	63.64%
PHE106-Main	GLU340-Side	38.00%	51.75%
LYS356-Side	GLU108-Side	26.22%	42.36%
THR345-Main	SER109-Main	37.56%	42.96%
GLY107-Main	GLU340-Side	39.33%	17.88%
ASN343-Side	TYR100-Side	41.33%	2.90%

**Table 3 ijms-24-00481-t003:** Predicted scoring of rosetta residue mutations.

Initial Residues	Predicted Residues	Energy Changes (kcal/mol)
P28	W	−14.5437
D99	T	−40.779
Y100	W	−12.019
G103	A	−4.81267
A104	T	−2.526
W105	H	−0.075
G107	H	−2.59663
L111	T	−2.15933

**Table 4 ijms-24-00481-t004:** Binding free energy of wild-type and mutant systems.

Systems	ΔG*bind* (kcal/mol)
WT	−54.7977
Y100W	−47.0290
D99T	−61.5002
G103A	−64.7284
P28W	−60.6685

## Data Availability

The complex of SARS-CoV-2 with antibody S309 is available in the PDB database (PDB ID:6wpt).

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
