# Peer review of "High-Affinity Antibodies Designing of SARS-CoV-2 Based on Molecular Dynamics Simulations"

_ijms, 2022, doi:10.3390/ijms24010481_

Round 1

Reviewer 1 Report

The work titled “High-Affinity Antibodies Designing of SARS-CoV-2 Based on

Molecular Dynamics Simulations” submitted by the authors is a very good in silico design of SARS-CoV-2 antibodies. In the given context, this could be a very good reference study for future antibody design against any known pathogen. The molecular dynamic simulation was performed well. I suggest to publish this work.

Author Response

Thanks for the reviewer's comments. Please see the response in the attachment.

Reviewer 2 Report

·      English should be improved.

·      Line 52-53 “as well as the high infection rate lead to the risky of physiological experiments” is not clear to the reader.

·      Add an image showing the hydrogen bonding discussed in line 91-97 and 119-128.

·      Figure 8 has legend error.

·      Conclusion typing error

Author Response

Thanks for the reviewer's comments. Please see the reponse in the attachment.
